# Exogenous Kinetin Promotes the Nonenzymatic Antioxidant System and Photosynthetic Activity of Coffee (*Coffea arabica* L.) Plants Under Cold Stress Conditions

**DOI:** 10.3390/plants9020281

**Published:** 2020-02-21

**Authors:** Robert Acidri, Yumiko Sawai, Yuko Sugimoto, Takuo Handa, Daisuke Sasagawa, Tsugiyaki Masunaga, Sadahiro Yamamoto, Eiji Nishihara

**Affiliations:** 1The United Graduate School of Agricultural Sciences, Tottori University, 4-01 Koyama-cho Minami, Tottori 680-8553, Japan; acidrirobert24@gmail.com (R.A.); actias.luna0530@gmail.com (T.H.); ssgw.2653.02.06@gmail.com (D.S.); 2Sawai Coffee Limited, 278-6, Takenouchi danchi, Sakaiminato City, Tottori 648-0046, Japan; yumiko@sawaicoffee.co.jp; 3Tottori Institute of Industrial Technology, 2032-3, Nakano-cho, Sakaiminato-shi, Tottori 684-0041, Japan; 4Faculty of Soil Eco-engineering and Plant Nutrition, Shimane University, 1060, Nishikawatsucho, Matsue 690-8504, Japan; masunaga@life.shimane-u.ac.jp; 5Faculty of Agriculture, Tottori University, 4-101 Koyama-cho Minami, Tottori 680-8553, Japan; yamasada@tottori-u.ac.jp

**Keywords:** chlorophyll fluorescence, cold stress, gas exchange, kinetin, membrane stability, antioxidant compounds, radical scavenging capacity

## Abstract

Coffee plants are seasonally exposed to low chilling temperatures in many coffee-producing regions. In this study, we investigated the ameliorative effects of kinetin—a cytokinin elicitor compound on the nonenzymatic antioxidants and the photosynthetic physiology of young coffee plants subjected to cold stress conditions. Although net CO_2_ assimilation rates were not significantly affected amongst the treatments, the subjection of coffee plants to cold stress conditions caused low gas exchanges and photosynthetic efficiency, which was accompanied by membrane disintegration and the breakdown of chlorophyll pigments. Kinetin treatment, on the other hand, maintained a higher intercellular-to-ambient CO_2_ concentration ratio with concomitant improvement in stomatal conductance and mesophyll efficiency. Moreover, the leaves of kinetin-treated plants maintained slightly higher photochemical quenching (qP) and open photosystem II centers (qL), which was accompanied by higher electron transfer rates (ETRs) compared to their non-treated counterparts under cold stress conditions. The exogenous foliar application of kinetin also stimulated the metabolism of caffeine, trigonelline, 5-caffeoylquinic acid, mangiferin, anthocyanins and total phenolic content. The contents of these nonenzymatic antioxidants were highest under cold stress conditions in kinetin-treated plants than during optimal conditions. Our results further indicated that the exogenous application of kinetin increased the total radical scavenging capacity of coffee plants. Therefore, the exogenous application of kinetin has the potential to reinforce antioxidant capacity, as well as modulate the decline in photosynthetic productivity resulting in improved tolerance under cold stress conditions.

## 1. Introduction

The genus *Coffea* comprises of over 126 species, all of which evolved as understorey species in the tropical rainforests of continental Africa, Madagascar and the Mascarene islands [1]. Out of these, two species (*C. canephora* Pierre ex A. Froehner and *C. arabica* L.) have been commercialized and subsequently cultivated in many countries, such as Brazil, which is the leading producer of coffee globally [2]. In many of the regions where the plants have been introduced, low, non-freezing temperatures during the winter season are common. Coffee species are very sensitive to chilling, which not only affects vegetative growth, but causes substantial yield losses [3].

Low temperatures (below 18 °C) cause a marked decrease in the photosynthetic rate, which is induced by both biochemical and diffusive limitations [3,4,5]. The low stomatal conductance and the corresponding low internal to ambient carbon dioxide concentration ratio have been reported to cause low photosynthetic rates resulting from low carboxylation and/or the mesophyll efficiency of CO_2_ fixation capacity. In addition to the repression of several carboxylation enzymes, low temperatures are associated with a rapid breakdown of photosynthetic pigments, such as chlorophylls, hence causing a positive feedback loop, which further reduces the photosynthetic rates. The resultant low CO_2_ fixation capacity causes higher degrees of photoinhibition resulting from excessive excitation energy, whose chances of occurrence are higher when the photosynthetic apparatus is stressed [6]. The subsequent photooxidative stress is associated with an overproduction of reactive oxygen species (ROS), as well as charged states of chlorophylls, namely ^3^Chl * and ^1^Chl in the chloroplasts [7]. However, leaves contain carotenoids (molecules), which dissipate excess excitation energy away from the PSII through de-epoxidation of the xanthophyll cycle in which violaxanthin is converted to xanthophyll, hence avoiding the generation of ROS [6,8,9]. Nevertheless, this generation of ROS is not only restricted to chloroplasts, but rather occurs in the mitochondria and the peroxisomes due to presence of several oxidative and electron transport reactions in these organelles [10,11]. Although ROS are produced under normal cellular metabolism as signaling molecules, their exacerbated production under stress conditions causes the oxidation and disintegration of lipids, proteins, pigments and DNA, as well as the inactivation of the enzymes of the photosystems and might culminate in cell death [12,13].

Plants are endowed with an elaborate antioxidant mechanism, which is comprised of both enzymatic and nonenzymatic antioxidants systems. During oxidative stress conditions, plants upregulate the activities of several enzymes, such as superoxide dismutase, catalase and peroxidases, which scavenge the ROS [14]. In addition, plants rely on the nonenzymatic, antioxidant molecules which can be either lipophilic or hydrophilic for the effective scavenging of the reactive radicals. Recently, it has been reported that the employment of either nonenzymatic or enzymatic antioxidative systems depends upon the abiotic stress factor to which the plants are exposed. That is, the superimposition of both cold and drought stress simultaneously reinforced the two antioxidative systems, while the enzymatic antioxidative systems was more reinforced during single exposure to drought than to cold stress. On the other hand, exposure to cold stress resulted in the accumulation of antioxidant compounds with less effect on the activities of the antioxidant enzymes [15]. Therefore, during cold stress conditions, while the nonenzymatic antioxidant system is more repressed, coffee plants rely on an alternative antioxidant system, which is made up of antioxidant compounds. It has been reported that young coffee leaves accumulate high amounts of phenolic compounds, such as chlorogenic acids, mangiferin and other phenolic compounds for ROS scavenging, rather than on enzymatic antioxidants [16]. The accumulation of phenolic compounds, such as anthocyanins, is also associated with an increased adaptation to abiotic stress conditions [17,18]. Moreover, coffee plants also contain high amounts of alkaloids, mainly caffeine and trigonelline, with potential antioxidant properties [19,20]. The antioxidant potency of these compounds and their related secondary metabolites is attributed to the presence of hydroxyl components and glycosylic linkages (Appendix A) which scavenge the ROS [21].

The accumulation of these compounds, in particular polyphenols, is associated with increased ROS scavenging capacity, which is normally assessed as vitamin E equivalent and expressed as trolox (tocopherol analogue) equivalent antioxidant capacity [22,23].

The exogenous foliar application of plant growth hormones is recently being explored as a possible means of mitigating, as well as conferring tolerance, to abiotic stress conditions [24,25,26,27,28,29]. Substances such as cytokinins are well known phytohormones that promote cell division and differentiation [30]. Despite their role in mediating several biological processes during normal cellular metabolism and development, as well as the prevention of oxidation during abiotic stress conditions, several reports have indicated a decline in the endogenous levels of cytokinins during oxidative stress conditions [31,32]. The exogenous supplementation of such phytohormones might be useful in maintaining normal cellular metabolism during abiotic stress conditions. Amongst the cytokinins, kinetin (N ^6^-furfuryladenine), a compound first isolated from autoclaved herring sperm DNA [33], recently received tremendous attention for its oxidative stress mitigation and growth promotion effects [34,35,36]. Moreover, due to its abundance amongst the cytokinins, its commercial applications, particularly in agricultural science, has been implored [37]. The exogenous foliar application of kinetin with concentrations ranging from 0.01 mM to 2 mM has been reported to increase the growth rate, yield, crop quality and tolerance to abiotic stress conditions, such as salinity, heavy metal stress and drought, among others in several crops [29,34,36,37,38]. Although the mechanism of kinetin action in mitigating oxidative stresses is not yet clearly understood, it has been suggested that kinetin upregulates both the enzymatic and the nonenzymatic antioxidative systems, hence reinforcing the ROS scavenging abilities of plants [34]. The exogenous foliar application of kinetin activates and stimulates the biosynthesis of secondary metabolites through the upregulation of the corresponding transcripts or by increasing the activities of the biosynthetic enzymes in a number of plants [37,39,40]. Therefore, due to the repressed enzyme activity during cold stress conditions, amplification of the nonenzymatic antioxidative system through kinetin application might offer a necessary tolerance mechanism to enhance the growth of plants, such as coffee.

Of the two commercially important species of coffee, *C. arabica* is the most significant, contributing over 60% on the global market. This species is relatively more tolerant to cold stress conditions, due to its ability to maintain a higher photochemical efficiency of PSII, as well as the triggering of several biochemical adjustments, unlike its counterpart *C. canephora*. Consequently, *C. arabica* is more grown in highland areas, where chilling temperatures are common during winter conditions. Moreover, the negative impacts of cold stress on the growth and productivity of coffee crops is likely to be exacerbated by current trends in climatic changes. In this study, we evaluated the potential of the exogenous foliar application of kinetin on the reinforcement of the nonenzymatic oxidative system of young *C. arabica* plants towards the improvement of photosynthetic physiology and the tolerance of coffee plants to cold stress conditions.

## 2. Results

### 2.1. Gas Exchange

Although the net CO_2_ assimilation rate was not significantly different amongst the treatments, possibly due to relatively large differences between the individual measurements for each coffee plant, exposure to cold stress conditions generally caused low diffusive gas exchanges while increasing both the intrinsic and the instantaneous water use efficiencies (Table 1). The high values of water-use efficiencies were as a result of significant reductions in stomatal conductance and transpiration rates, whose decline is normally associated with low photosynthetic rates. The cold-induced decline in stomatal conductance and the transpiration rate was, however, more apparent in non-kinetin-treated plants (79.6% and 77.3%) than in the kinetin-treated plants (23.4% and −2.27%), respectively. Moreover, under cold stress conditions, kinetin treatment caused an improvement in the net CO_2_ assimilation rate by 63.8%.

This was accompanied by higher stomatal conductance (121.7%), transpiration rate (111.8%) and intracellular CO_2_ concentration (99.4%), as well as a higher internal-to-ambient CO_2_ concentration ratio (87.8%) in the kinetin compared to non-treated kinetin counterparts under cold stress conditions. Similarly, the mesophyll efficiency for carbon fixation was 18.7% higher in kinetin-treated plants under cold stress conditions than in the non-kinetin-treated plants. This means that the exogenous foliar application of kinetin tended to reduce the negative effects induced by cold stress conditions in coffee plants, by maintaining relatively higher gas exchange rates in the treated plants compared to the non-treated plants.

### 2.2. Chlorophyll Fluorescence

Similar to gas exchange, cold stress conditions reduced the photosystem II photochemical efficiency more in non-kinetin-treated plants than in the kinetin-treated plants (Table 2). This was evinced by a larger reduction in the photosystem II operating efficiency (Φ_PSII_) in the non-kinetin-treated plants (40.0%), compared to (30.8%) in kinetin-treated plants on exposure to cold stress conditions. This was also accompanied by similar variation in the electron transfer rate, which diminished by 38.0% and 30.3% in the two groups, respectively. Despite the proportion of light energy allocated to photochemistry (qP), and that dissipated as heat (photochemical and nonphotochemical quenching, respectively), not showing significant variation amongst the treatment, the latter (NPQ) showed a remarkable reduction with cold stress conditions more in the non-kinetin-treated plants 46.5%, compared to 36.6% in the kinetin plants, respectively. Similarly, fluorescence quenching (qN) was lowest under cold stress conditions, which was also associated with the cold-stress-induced closure of photosystem reaction centers (qL), especially in the non-kinetin-treated plants (67.0%), compared to the kinetin-treated plants, which maintained 0.35% of the PSII reaction centers open during cold stress conditions. The current results suggest that this exogenous kinetin treatment caused no significant improvement in the chlorophyll fluorescence parameters in the leaves of coffee plants under cold stress conditions after 5 days. Nevertheless, long-term assessments exposure might result in improvements of chlorophyll fluorescence in the kinetin-treated plants compared to their non-treated counterparts under cold stress conditions.

### 2.3. PSII quantum Efficiency

The quantum efficiency of PSII evaluated as the ratio of variable to maximum and variable to initial fluorescence yield indicated increased sensitivity to photoinhibition in cold stressed leaves (Table 3). Although no improvement in the PSII quantum efficiency was observed in kinetin-treated plants under cold stress conditions, the exogenous foliar application of kinetin slightly improved the effective quantum efficiency during the light-adapted state, as indicated by higher amplitudes of both Fv’/Fm’ and Fv’/Fo’ in kinetin treatment plants; 0.56 and 1.30, compared to 0.52 and 1.08, respectively, in control treatment plants under optimum conditions. Therefore, the current results indicate that the expected positive effects of the application under the conditions on PSII quantum efficiency might be realized in a longer term, rather than over a short period of time.

### 2.4. Electrolyte Leakage Index

Membrane disintegration as measured by the electrolyte index indicated that exposure to cold stress conditions caused injury to the cell membrane (Figure 1). Under both optimum and cold stress conditions, the exogenous foliar application of kinetin increased the leakage of cellular electrolytes by 39.0% and 33.4%, compared to control and cold-treated plants, respectively. In fact, the electrolyte index was the highest at 15.7% in the kinetin-treated plants under the stress conditions compared to 8.24%, 11.4% and 11.7% in control, kinetin and cold treated plants, respectively. Contrary to the expected, the current results indicate that the exogenous application of kinetin tended to increase the content of ROS, which was accompanied by a higher electrolyte leakage index. Nevertheless, this could be a short-term effect, and therefore long-term exposure might yield contrasting results.

### 2.5. Photosynthetic Pigments

The content of individual chlorophyll types *a* and *b* and their total (*a* + *b*) decreased on exposure to cold stress conditions, as well as on the exogenous foliar application of kinetin by 14.1%, 16.8% and 21.2% for Chl *a* and 17.7%, 21.3% and 24.4% for Chl *b* and 15.0%, 17.9% and 22.1% for total chlorophylls in kinetin, cold and cold + kin treatments, respectively compared to control-treated plants (Figure 2A–C, respectively). On the contrary, the content of carotenoids was stimulated by the foliar application of kinetin under optimum conditions, hence reaching a maximum content of 1.65 mg gDW^−1^ compared to 1.28 mg gDW^−1^ in control-treated plants (Figure 2D). Nevertheless, this promotive effect was not observed under cold stress conditions, since the content of carotenoids was non-significantly different between cold and cold + kin treated plants at 1.43 and 1.35 mg gDW^−1^, respectively. Although the ratio of chlorophyll *a* to *b* was not significantly affected by either cold stress imposition or kinetin treatments (Figure 2E), that of chlorophylls to carotenoids (Chl/Car) was significantly reduced by 33.2, 26.7 and 26.4 in kinetin, cold and cold + kin treated, compared to control-treated plants (Figure 2F). It is suggested therefore that, despite increasing the content of carotenoids during optimum conditions, the protective role of kinetin on chlorophyll pigments could not be realised after 5 days of cold stress conditions, although contrasting results may be obtained in long-term studies.

### 2.6. (HPLC) Nonenzymatic Antioxidants Compounds

Under both optimal and cold stress conditions, kinetin treatment caused a consistent upsurge in the content of the small weight compounds determined by HPLC (Figure 3). On average, the content of caffeine, trigonelline, 5-caffeoylquinic acid and mangiferin were increased by 64.3%, 50.0%, 96.0% and 71.4%, respectively, under optimum conditions, hence causing an increase of 185% in the total HPLC-determined compounds as a result of kinetin treatment (Figure 3A–F, respectively).

Similarly, under cold conditions, kinetin elicited an increase of 20.5%, 23.7%, 27.1% and 22.2% in the contents of caffeine, trigonelline, 5-CQA and mangiferin, respectively, and an increase of 25.8% in the total content of the HPLC compounds (Figure 3A–F, respectively). Although the contents of both the alkaloids and phenolics were upregulated by kinetin treatment, the magnitude of increment was much more for the latter than for the former. As a result, the ratio of 5-CQA/Caffeine was increased by 20.6% and 5.3% under optimum and cold stress conditions, respectively (Figure 3E). Nonetheless, in the absence of kinetin, cold stress conditions also tended to increase the content of the antioxidants in the non-kinetin-treated plants. Compared to optimum conditions, exposure to cold stress conditions increased the contents of caffeine, trigonelline, 5-CQA and mangiferin by 33.3%, 35.7%, 56.7% and 28.5%, respectively, causing a total upsurge of 116.3% in the HPLC antioxidants (Figure 3A–F, respectively). Similar to kinetin treatment, cold treatment elicited more 5-CQA than caffeine, resulting in an increment of 20.3% compared to optimum conditions (Figure 3E). Nevertheless, the current results indicate that the exogenous foliar application of kinetin stimulated the metabolism of both alkaloid and phenolic compounds that were determined by the HPLC, hence causing their accumulation in the treated plants.

### 2.7. Total Anthocyanin Content (TAC)

Accumulation of anthocyanins in coffee leaves was expressed on the basis of both peonidin-3-(6-caffeoyl-sophoride)-glucoside and cyanidin-3-glucoside equivalents (Figure 4A,B, respectively). Neither kinetin nor cold treatments significantly affected TAC in coffee leaves (Figure 4). Nevertheless, TAC was consistently the highest in kinetin treatment under both cold stress and optimum temperature conditions at 1.74 and 1.61 mg YGM-5b eq gDW^−1^ and 0.84 and 0.76 mg Cy3G eq gDW^−1^, respectively (Figure 4A,B). On the other hand, TAC was lowest in control treatment plants at 1.15 YGM-5b eq gDW^−1^ and 0.56 mg Cy3G eq gDW^−1^, while cold stress conditions had less effect on the content of the anthocyanins with cold-treated plants, containing 1.39 mg YGM-5b eq gDW^−1^ and/or 0.66 mg Cy3G eq gDW^−1^. Despite varying non-significantly amongst the treatments, the results show that kinetin treatment tended to increase the accumulation of anthocyanin under both optimum and cold stress conditions.

### 2.8 Total Phenolic Content (TPC)

TPC showed a typical trend in variation amongst the treatments, being at the highest in kinetin-treated plants at 55.5 and 54.9 mg gDW^−1^ GAE under optimum and conditions, respectively (Figure 5). Contrastingly, control treatment plants accumulated the least TPC (38.9 mg gDW^−1^ GAE), while exposure to cold stress conditions slightly increased the TPC to 42.4 mg gDW^−1^ GAE (Figure 5). These results indicate that the exogenous foliar application of kinetin elicited an accumulation of phenolic compounds in the leaves of the coffee plants

### 2.9. Radical Scavenging Capacity

Like the variations in the content of the analyzed compounds, the radical scavenging capacity as expressed by the TEAC values was consistently highest in kinetin-treated plants under the two temperature conditions (Table 4). The FRAP assay indicated that the radical scavenging capacity was significantly highest in kinetin-treated plants under optimum conditions, followed by those under cold stress conditions at 1138.6 and 938.7 µmoL Trolox gDW^−1^, respectively. Similarly, both DPPH and ABTS TEAC values were high at 317.9 and 277.2 µmoL Trolox gDW^-1^ in kinetin treatment and, 270.1 and 281.6 µmoL Trolox gDW^−1^ for cold + kin treatment plants, respectively. On the other hand, control and cold-treated plants had low radical scavenging capacity values with the assays, indicating TEAC values of 752.5, 262.6 and 211.3 µmoL Trolox gDW^−1^ in control plants, and 542.1, 236.9 and 190.2 µmoL Trolox g^−1^ in cold-stressed plants for FRAP, ABTS and DPPH assays, respectively (Table 4). Consequently, the DPPH IC_50_ was at the least in the kinetin-treated plants at 114.4 and 102.8 µgDW mL^−1^ under both optimum and cold stress conditions, respectively, while non-kinetin-treated plants indicated the highest values at 125.8 and 154.0µgDW mL^−1^ for the two respective temperature conditions (Table 4). It is suggested therefore that this kinetin treatment increased the ability of coffee leaves to scavenge the ROS which are more detrimental during oxidative stresses, typical of cold conditions.

## 3. Discussion

Temperatures below 18 °C are one of the abiotic stresses limiting the productivity and growth of coffee plants [41]. Abiotic stresses including cold stress reduce the CO_2_ fixation capacity, which is normally associated with a low net CO_2_ assimilation rate, stomatal conductance and transpiration rate in plants [8]. Subjecting coffee plants to 12 °C in the current study induced cold stress conditions which were associated with low gas exchanges, hence inducing diffusive limitations to photosynthetic productivity in coffee plants (Table 1). Similar effects of cold stress conditions on the photosynthetic physiology of coffee plants have been reported [41,42]. These limitations are accompanied by a reduction in the PSII operating efficiency, resulting from the breakdown of the photosynthetic apparatus under cold stress conditions [43]. This is in agreement with the observations made in the current study, where cold-stressed plants had the least values of Φ_PSII_, which was associated with low electron transport rates (Table 2). The reduction in the CO_2_ fixation capacity, together with low PSII operating efficiency, result in an over production of reactive oxygen from any given incident light [44]. This normally results in a phenomenon known as photoinhibition, when excess amounts of ROS are produced in the antenna pigments of the chloroplasts by the Mehler reaction [12,45]. An overproduction of ROS in the photosynthetic apparatus typical under oxidative stress conditions causes a severe reduction in both the effective and maximum quantum yield of PSII [6]. This is in agreement with the results of the current study, where cold-stressed plants had the low values of quantum yield in the dark and light-adapted chlorophyll states (Table 3). Nevertheless, the exogenous application of kinetin tended to modulate the dampening effects of cold stress on the PSII operating efficiency, hence resulting in improved gas exchange, which was accompanied with higher Φ_PSII_, ETR and an improved quantum efficiency in kinetin-treated plants compared to the non-treated plants under cold stress conditions (Table 1, Table 2 and Table 3). Our results agree with a number of studies which reported the positive effects of exogenous kinetin treatment on the photosynthetic machinery [34,46]. Although the protective roles of kinetin are concentration dependent, similar to the concentration used in the current study, de Moura et al. [46] reported that 0.35 mM (75 mg L^−1^) effectively improved the carboxylation efficiency, with corresponding effects on the photosynthetic apparatus. It has been reported that kinetin induces stomatal opening, hence encouraging gas exchange with associated net CO_2_ assimilation under stress conditions [47]. This is associated with both the protective effects of kinetin on the functioning of the photosynthetic light reaction and the functional enzymes of PSII and PSI, such as the protein gradient regulation-5 (PGR-5) protein, as well as maintaining a pH gradient for a smooth electron flow from PSII to PSI [48].

Our results further indicated that exposure to cold stress conditions had a negative impact on the membrane stability in the leaves of the coffee plants (Figure 1). This could be attributed to an increased production of ROS in the chloroplasts, as well as other organelles with an electron chain system, such as the mitochondria and peroxisomes [49]. ROS are normally produced by plants under optimal conditions, in this case acting as signaling and transduction molecules for normal cellular metabolism [10,13]. However, their amplified production during oxidative stress conditions results in a disruption of the normal cellular homeostasis, leading to membrane lipid peroxidation characterized by the leakage of cellular electrolytes, protein oxidation and enzyme inhibition, as well as the disintegration of deoxyribonucleic acid and ribonucleic acid [10,50]. Cold stress conditions also resulted in the breakdown of chlorophyll molecules (Figure 2). The accumulation of high amounts of ROS in the photosynthetic machinery particularly breaks down the D1 protein subunit of the PSII reaction center, leading to the oxidation and bleaching of chlorophyll molecules [7,51]. Exogenous kinetin applications in several concentrations have been reported to stabilize cellular membranes in several crops exposed to a heavy metal stress [52,53], waterlogged conditions [53], UV-B stress [35] and salinity stress [34], which is associated with high amounts of photosynthetic pigments resulting from kinetin-mediated enhancement in the expression of genes that encode several proteins involved in pigment biosynthesis [54]. In the current study, besides increasing carotenoid content under optimum conditions, kinetin application elicited contrasting results to the prior mentioned studies. Kinetin application tended to increase membrane damage, as well as the disintegration of chlorophyll molecules (Figure 1 and Figure 2, respectively). This could be related to the kinetin mechanism of action after exogenous application, through which it increases nicotinamide adenine dinucleotide activity, leading to the generation of H_2_O_2_, which acts as a signaling molecule for antioxidant defense [54]. Although the short-term increase in the content of the ROS is associated with increased cellular damage, it might substantially contribute to the overall enhancement of the antioxidant defense system, with improved tolerance to abiotic stress conditions in the long run. On the other hand, the up-regulatory effect of kinetin on carotenoids metabolism also coincides with the metabolism of polysaccharides and proteins of the photosynthetic apparatus, and this in fact causes the stimulation of the entire biogenesis of chloroplasts [55,56].

The kinetin–hydrogen peroxide-mediated reinforcement of the antioxidant system is supported by high amounts of antioxidant compounds analyzed in the current study (Figure 3, Figure 4 and Figure 5). The current results are in agreement with several reports were exogenous kinetin treatment increased the tolerance of plants to abiotic stress conditions through the increased activities of antioxidant enzymes [34,35,36,52], as well as the increased contents of several antioxidant compounds, such as flavonoids, gibberellins, salicylic acid, jasmonic acid and abscisic acid in salinity-stressed soybean plants [37], proline in drought-stressed rice [39], salvianolic acid and rosmarinic acid in *Dracocephalum forrestii* [40]. Although the accumulation of several secondary metabolites in the leaves of coffee plants under oxidative stress conditions is a well-known phenomenon [15,16], this study is the first to report a profound increment in the contents of these compounds in response to exogenous kinetin treatment under optimal or cold stress conditions. These secondary metabolites play a number of roles in the plant defense mechanisms against oxidative stress due to the presence of hydroxyl components and glycosylic linkages which scavenge the ROS, hence maintaining cellular homeostasis during oxidative stress conditions [21,57,58,59]. Kinetin increased the content of both alkaloids, such as trigonelline and caffeine, and phenolic compounds, including 5-caffeoylquinic acid, the main chlorogenic acid in coffee plants [60], mangiferin and anthocyanins. Unlike alkaloids such as caffeine, whose role in the detoxification of ROS is less reported [61], the accumulation of phenolic compounds has been associated with an increased tolerance to several abiotic stress conditions [15,16,17,18]. Nevertheless, trigonelline (1-methylpyridinium-3-carboxylate), another alkaloid compound, is a reservoir for NAD biosynthesis, and therefore its accumulation may contribute to the cellular energy metabolism which might lead to enhanced abiotic stress tolerance [61]. During cold stress conditions when the activities of the enzymatic antioxidants are repressed by low temperatures, coffee plants were found to rely on the accumulation of mainly phenolic compounds as alternative defense mechanisms against oxidative stress [15].

Accumulation of these powerful antioxidants is more important in the young leaves of coffee plants, whose enzymatic antioxidant system is normally underdeveloped at this stage, and therefore unable to neutralize the rapid accumulation of ROS from intense oxidative stresses [16]. Therefore, by increasing the content of these compounds, exogenous kinetin application improves the nonenzymatic antioxidant system of the young coffee plants. The current study agrees with a number of reports, where kinetin enhanced the production of phenolic and alkaloid compounds in the explants of different species under several abiotic stress conditions [62,63,64].

Although the mechanism through which exogenous kinetin influences the metabolic pathways of these compounds is not yet known, it has been suggested that kinetin upregulates the associated transcription factors [30], as well as directly enhancing the activities of phenolic and alkaloid biosynthesis enzymes [62,64]. This could be due to kinetin-induced increase in the concentration of H_2_O_2_ molecules, which although produced in the chloroplasts, they rapidly diffuse into the nucleus, where they act as signaling molecules for the metabolism of phenolic and alkaloid compounds, hence contributing to antioxidant defense against the overproduction of ROS [65]. In addition, it has been reported that kinetin scavenges free oxygen radicals by directly neutralizing ROS using the hydrogen from the α-carbon of the amine bond of N^6^-furfuryladenine. Kinetin also reacts with copper, forming a Cu (II)-kinetin complex, which encourages a faster dismutation of the radical oxygen species in solution [30], hence contributing to the antioxidative capacity of plants. Taken together, the exogenous foliar application of kinetin increased the ROS scavenging capacities in the treated plants (Table 4). This increase was more ostensibly indicated by the FRAP assay, whereas DPPH and ABTS, despite showing increment tendencies with kinetin treatment, were not affected significantly amongst the treatments. Although the assays employ different principles in the determination of radical scavenging values, they normally report consistent values [22,23,66], as was observed in the current study.

## 4. Materials and Methods

### 4.1. Plant Material and Growing Conditions

Coffee (*Coffea arabica* L.) seeds were imported from Bali, Indonesia, and provided by the Sawai Coffee Company, Limited (Tottori, Japan). The seeds had their seed coat removed, were soaked in running water for 3 days, and thereafter germinated on a moist paper towel in an incubator in the dark at 30 °C for 14 days [67]. The sprouted seeds after 21 days were then sown in a soil culture consisting of peat moss, perlite and humus, at a ratio of 5:3:2, and contained in individual pots of 10 cm diameter. The seedlings were grown in a greenhouse, whose temperatures were maintained above 20 °C for one year. During growth, fertigation was done with a half strength nutrient solution, whose composition was adopted from Hoagland and Arnon [68], with modifications. The nutrient solution consisted of, in mmol L^−1^, 2.9 N-NO_3_^−^, 0.5 N-NH_4_^+^, 0.05 P-H_2_PO_4_^-^, 1.2 Ca^2+^, 0.3 Mg^2+^ and 0.4 S-SO_4_^2−^, and in µmoL L^−1^, 17.5 Fe (III) EDTA, 0.4 Cu, 0.8 Zn, 3 Mn, 9 B and 0.05 Mo, which was applied depending on the soil culture condition. Coffee seedlings were cultivated until they reached 5-pairs leaf stage. After that, for experimentation, uniform-sized seedlings were selected and acclimated in a growth chamber for 3 months under optimum, ambient environmental conditions; day and night temperature of 25/20 °C, respectively, relative humidity of 70% and photosynthetic photon flux density (PPFD) of 250 µmoL m^−2^ s^−1^ provided by white fluorescent tubes with a 12 h photoperiod.

### 4.2. Kinetin and Cold Stress Treatments

After acclimation, the plants were randomly assigned to two temperature conditions and two kinetin treatments in a 2 × 2 factorial experimental design. The four treatments were as follows: (1) control, for which the plants were maintained at optimum temperatures (day/night, 25/20 °C, respectively), and sprayed with distilled water; (2) Kinetin, sprayed with 0.35 mM kinetin and maintained at optimum temperature; (3) Cold, sprayed with distilled water and subjected to cold stress; (4) Cold + Kin, sprayed with 0.35 mM kinetin and subjected to cold stress.

Basing on related works [29,34,39,46], it was presumed that the concentration of kinetin used in the current study would have significant physiological effects on the coffee plants. Foliar sprays were administered three times every after 3 days, using a hand sprayer immediately after the growth chamber lights were routinely turned off. In all of the treatments, 0.1% isopropyl alcohol (Kao Global Chemicals, Tokyo, Japan) surfactant was used. Cold stress conditions were started 24 h after the last foliar sprays by reducing the growth chamber temperature from 25/20 °C to 12/12 °C, day/night, respectively, for 5 days, while maintaining the other ambient conditions constant.

### 4.3. Photosynthetic Measurements

Gas exchange and chlorophyll fluorescence parameters were measured using a portable open-flow gas exchange system (LiCOR 640°0XT, LiCOR, NE, USA). The measurements were done during both optimal conditions, and at 5 days after instigating cold stress conditions at the sixth hour of illumination, which corresponded to midday. Both gas exchange measurements and the light-adapted state of chlorophyll *a* fluorescence were simultaneously measured during the light-adapted state. The dark-adapted state measurements were done six hours after setting darkness when the growth chamber lights were turned off. Before each measurement day, the system was checked for leaks, and calibrated at a CO_2_ concentration of 400 ppm and light intensity of 500 µmoL m^−2^ s^−1^ PPFD provided by blue/red light-emitting diode (LED). Chlorophyll-*a* fluorescence parameters were thereafter calculated according to Murchie and Lawson [69], whereas the instantaneous and the intrinsic water-use efficiency, as well as the mesophyll efficiency, were calculated as the ratios of the net CO_2_ assimilation to the transpiration rate, (A/E); the net CO_2_ assimilation rate to stomatal conductance, (A/g_s_), and the intracellular CO_2_ concentration to stomatal conductance (C_i_/g_s_), respectively.

### 4.4. Electrolyte Leakage Index Determination

Membrane damage in the coffee leaves was assessed by electrolyte leakage index (ELI), using an electrical conductivity meter (Laqua F-74, Horiba, Japan), according to Zhu et al. [70]. Fresh leaf tissues were washed thoroughly with distilled water to remove surface contamination, cut in to 1 cm strips and placed in 15 mL stoppered tubes containing 10 ml of ultrapure water. The samples were then shaken at 100 rpm on a shaker at room temperature (25 °C) for 24 hrs. The electrical conductivity (EC) of the bathing solution was recorded as EC1. Thereafter, the tubes with their contents were placed into boiling water for 15 minutes, and then cooled to room temperature, after which their new EC was recorded as EC2. The electrolyte leakage index was calculated as EC1/EC2, and was expressed as a percentage.

### 4.5. Biochemical Assays

Leaves for biochemical analysis were collected at the sixth hour of illumination (corresponding to midday) at the end of the experiment. The samples were immediately flash frozen in liquid nitrogen and stored under -80 °C for further analysis. For all the assays, the samples were freeze-dried at -20 °C (Eyela DRC 1000-FDU 1110, Tokyo, Japan), and thereafter milled into a powder using a blender (Wonder Blender Osaka, Japan).

#### 4.5.1. Photosynthetic Pigments

Chlorophylls and carotenoids were extracted from 50 mg of the freeze-dried powder using 80% chilled acetone in 50 mL Falcon conical centrifuge tubes (Thermo Fisher Scientific, Massachusetts, USA) under dark conditions. The mixture was sonicated in an ice-cold water bath and centrifuged at 29,300×g at a temperature of 4 °C (Hitachi high-speed refrigerated centrifuge CR21N, Hitachi Koki Co. ltd, Tokyo, Japan). This extraction was repeated three times until all the green color from the residue was extracted. The supernatants for each sample were pooled together for analysis.

Quantification was done using a spectrophotometer (Hitachi ratio beam spectrophotometer, U-5100, Japan) at A_663.6_, A_646.6_, A_470.0_ and A_750.0_ nm, as according to Porra et al. [71]. Chlorophylls (a + b) and carotenoid (xanthophylls + β-carotene, x + c) contents were calculated according to equations in Porra et al. [71] and Lichtenthaler and Buschmann [72], respectively.

#### 4.5.2. Extraction and HPLC analysis of Selected Antioxidant Compounds

HPLC quantification of the selected antioxidant compounds was done according to Acidri et al. [22]. Two (2) alkaloids (caffeine and trigonelline) and two phenolic compounds (5-caffeoylquinic acid and mangiferin) were extracted and thereafter analyzed simultaneously from the freeze-dried powder samples in 50 mL Falcon tubes, using 80% methanol. Chlorogenic acids are the main phenolic compounds in coffee leaves, 5-CQA being the main isomer representing over 80% of the total hydroxycinnamic acid esters [60]. For, extraction, the mixture was sonicated and centrifuged at 29,300×g at a temperature of 10 °C. Extraction was repeated three times, and the supernatants for each sample pooled together and thereafter filtered (0.22 µm Millipore) for analysis.

Analysis was performed using a high-performance liquid chromatography (HPLC) system equipped with a UV detector (Hitachi L-2490, Hitachi, Japan) on a 10 µL sample extract. All the analytes were analyzed simultaneously using a linear gradient elution (Appendix A) in a reverse mode on a C18 column (TSKgel ODS-100, 4.6 × 150 mm, 5 µm made by Sigma-Aldrich, Tokyo, Japan) at 40 °C. The mobile phase consisted of filtered (0.22 µm Millipore), sonicated and degassed methanol (100%) and acetic acid solution (2%) solvents. Elution was done at 0.4 mL min^−1^ at a wavelength of 270 nm, resulting in a retention time of 4.64, 18.87, 20.27 and 22.35 for trigonelline, 5-CQA, caffeine and mangiferin, respectively (Appendix A). Analyte quantification was done by peak area measurement, and each compound determined linearly using regression equations developed from calibration curves for standard compounds. Calibration curves were obtained from three replicate points for each standard. All the analytical standards and the organic solvents were of HPLC grade (Sigma-Aldrich, Tokyo, Japan).

#### 4.5.3. Total Anthocyanin Content (TAC) Determination

Total anthocyanin content in the leaves was assayed according to Neff and Chory [73], with a few modifications. For each sample, 100 mg of the freeze-dried powder was incubated in 20 mL methanol, which was previously acidified with 1% HCl in a dark refrigerator at 4 °C for 24 h. The samples were then diluted with 10 mL of ultrapure water, and thereafter, the anthocyanins were separated from chlorophylls, using 20 mL of chloroform (triChloromethane). Spectrophotometric measurements were made at A_520.0_, A_530.0_, A_657.0_ and A_700.0_. Total anthocyanin content was then calculated as in [74], and expressed as both peonidin 3-(6-caffeoylsophoroside)-5 glucoside (YGM-5b) and cyanidin 3-glucoside (Cy3G) mg equivalents per unit dry mass.

#### 4.5.4. Total phenolic Content (TPC) Determination

Sample extraction and TPC determination was done as described by Acidri et al. [22]. 50 mg of freeze-dried leaf samples were infused in 25 mL boiling water in 50 mL falcon tubes for 8 minutes, and thereafter cooled under room temperature. The contents were then centrifuged at 29,300×g at 25 °C for 15 min. The supernatant was collected in 50 mL volumetric flasks, and to the residue was added 10 mL cold water for re-extraction two more times. For each sample, all the supernatants were pooled together, volumetric flasks filled to the mark. and thereafter filtered. using 0.45 µm filters in to 50 mL glass vials. From each sample, an aliquot of 2 mL was obtained for total phenolic content determination. while the rest was re-freeze dried at -20 °C for 7 days for radical scavenging capacity assays.

Total phenolic content was determined from 0.5 mL of the collected aliquots. Each sample was diluted 10-fold, from which 1 mL was transferred to a new test tube. The leaf extracts were then incubated with 5 mL of 10% Follin–Ciocalteu’s reagent for one minute and then reacted with 4 mL of 20% (*w/v*) sodium carbonate solution for 30 minutes at room temperature.

Gallic acid standards of serial concentrations prepared using methanol were similarly treated together with the samples. Absorbance was then read at 765 nm using a Hitachi ratio beam spectrophotometer (U-500 Hitachi, Tokyo, Japan).

#### 4.5.5. Radical Scavenging Capacity

Total radical scavenging capacities of the coffee leaves was assayed as described by Acidri et al. [22] using 5 mg of the freeze-dried, powdered sample obtained from the previous subsection. The samples were dissolved in 8 mL of ultrapure water and followed thereafter by methanol (100%) in 25 mL flasks. Each sample was then divided into 4 subsets of serial dilutions and stored under -80 ^°^C until further analysis.

##### 2,2-diphenyl-1-Picryl-Hydrazyl (DPPH) Radical Assay

The DPPH radical was used to determine the free radical scavenging capacity in coffee leaves. For analysis, the frozen samples were adjusted to room temperature, from which 1 mL was transferred to a new test tube, diluted 10-fold with methanol, and then incubated with freshly prepared 0.1 mmol L^−1^ DPPH solution for 10 minutes in the dark in a total volume of 10 mL. In addition, freshly prepared Trolox (6-hydroxy-2, 5, 7, 8-tetramethyl-chroman-2-carboxylic acid—a synthetic vitamin E analogue) standards of serial dilutions were treated in the same way as the samples. Absorbances of both the standards and samples were determined at 519 nm using the Hitachi ratio beam spectrophotometer. The concentration that caused a 50% decrease in the initial concentration of the DPPH radical defined as IC_50_, was determined for both the standard and samples from the percentage inhibition of the DPPH radical, which was calculated as % inhibition = (Abs_control_-Abs_sample_)/(Abs_control_-Abs_blank_) × 100, where Abs_control _ = absorbance of the 0.1 mmol L^−1^ DPPH methanol solution, Abs_sample _ = absorbance of the 0.1 mmol L^-1^ methanol solution after fading induced by addition of sample or Trolox, and Abs_blank_ = absorbance of the methanol solvent only. Antioxidant capacity of the samples as determined using the DPPH scavenging radical was calculated as (IC_50 (Trolox)_/IC_50 (sample)_) × 10^5^ and expressed as µmoL Trolox gDW^−1^ of Trolox equivalent antioxidant capacity (TEAC).

##### 2,2-Azino bis (3-Ethyl Benzothiazoline-6-Sulphonic Acid (ABTS) Radical Assay

For ABTS radical scavenging capacity, an ABTS solution of 7 mmol L^−1^ was prepared and mixed with a K_2_S_2_O_8_ solution of 140 mmol L^−1^ at a ratio of 5 mL:88 µL, respectively. The solution was then incubated at room temperature in the dark overnight. On the measurement day, the stock solution was diluted with ultrapure water at a ratio of 0.6:40 mL, respectively, and the absorbance adjusted to 0.7 ± 0.02 nm at 734 nm by spectrophotometry. Serial dilutions of freshly prepared Trolox standard and coffee leaf extracts of 0.5 mL were then incubated with 9.5 mL of the new solution for 10 minutes at room temperature in the dark, after which the absorbance was read at the same wavelength. ABTS antioxidant capacity was then calculated as gradient of Trolox standard/gradient of sample, and expressed as µmoL Trolox gDW^-1^ of TEAC.

##### Ferric Ion Reducing Antioxidant Power (FRAP) Assay

For the FRAP assay, the FRAP oxidant solution consisted of 0.2 M sodium acetate buffer (pH 3.6), 20 mM Ferric Chloride solution and 10 mM TPTZ (2,4,6 tris (2-pyridyl)-s-triazine) solution in 40 mM HCl mixed at a ratio of 10:1:1, respectively. The serially-diluted samples and Trolox standards of 0.5 mL were incubated with 9.5 mL of freshly prepared FRAP oxidant solution in a water bath at 37 °C for 40 minutes, after which the absorbance was read at 593 nm. The FRAP antioxidant capacity was then calculated as the gradient of Trolox standard/gradient of sample, and expressed as µmoL Trolox gDW^−1^ of TEAC.

### 4.6. Statistical Analysis

Experimental data were statistically examined using two-way analysis of variance (two-way ANOVA), following a completely randomized design for both kinetin and cold stress effects. The parameter differences were compared for statistical significance amongst the treatments using the Newman–Keuls test at *p* ≤ 0.05. Assumptions of normality were tested using the Kolmogorov–Smirnov test, and the data transformed accordingly to attain a normal distribution whenever necessary. All the statistical analyses were done using the Stata 12.0 statistics and data analysis program (Stata Corp, Texas USA). Data are expressed as means ± SD, *n* = 3.

## 5. Conclusions

Results of the current study indicated that exposure to cold stress conditions impacted the physiology and metabolic processes of coffee plants. The ROS generated as a result of oxidative stress associated with cold stress conditions caused the disintegration of the cell membranes and photosynthetic pigments. This was accompanied by diminished photosynthetic efficiency of the PSII with concomitant reductions in the gas exchange and photosynthetic activity in the leaves of coffee plants. On the other hand, the exogenous foliar application of kinetin improved the antioxidative capacity of the coffee plants by upregulating the metabolism of the nonenzymatic antioxidant compounds. This was associated with increased reactive species scavenging capacity in the kinetin-treated plants. Exogenous kinetin application slightly increased the photochemical and mesophyll efficiency for CO_2_ fixation in addition to maintaining somewhat higher gas exchanges, even under cold stress conditions. Therefore, on the basis of the results presented in the current study, it is suggested that kinetin has a potential to modulate the growth of the coffee plants under cold stress conditions, and therefore more studies should be conducted to explore its efficacy under field conditions. Exploring cold mitigation or tolerance enhancement is not only useful in the face of unpredictable climatic changes, but also is likely to expand coffee cultivation to new production areas, whose environmental conditions are currently unsuitable for coffee production.

## Figures and Tables

**Figure 1 plants-09-00281-f001:**
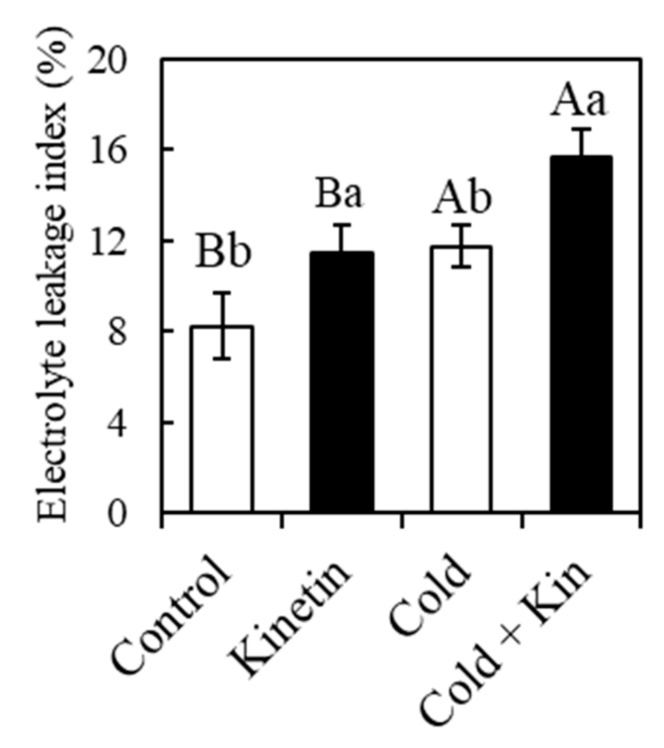
Effect of exogenous kinetin on the electrolyte leakage index in the leaves of coffee plants under optimum and cold stress conditions. Optimum temperature, 25/20 °C, whereas cold stress conditions, 12/12 °C. Kin, kinetin. Different capital letters, (A, B) denote statistically significant differences between the parameter means within temperature treatments, whereas different small letters, (a, b) denote statistically significant differences for the parameter means within kinetin treatments (*p* < 0.05, Newman–Keuls test). Means are a replicate of three independent samples (*n* = 3). Error bars represent standard deviation of the mean.

**Figure 2 plants-09-00281-f002:**
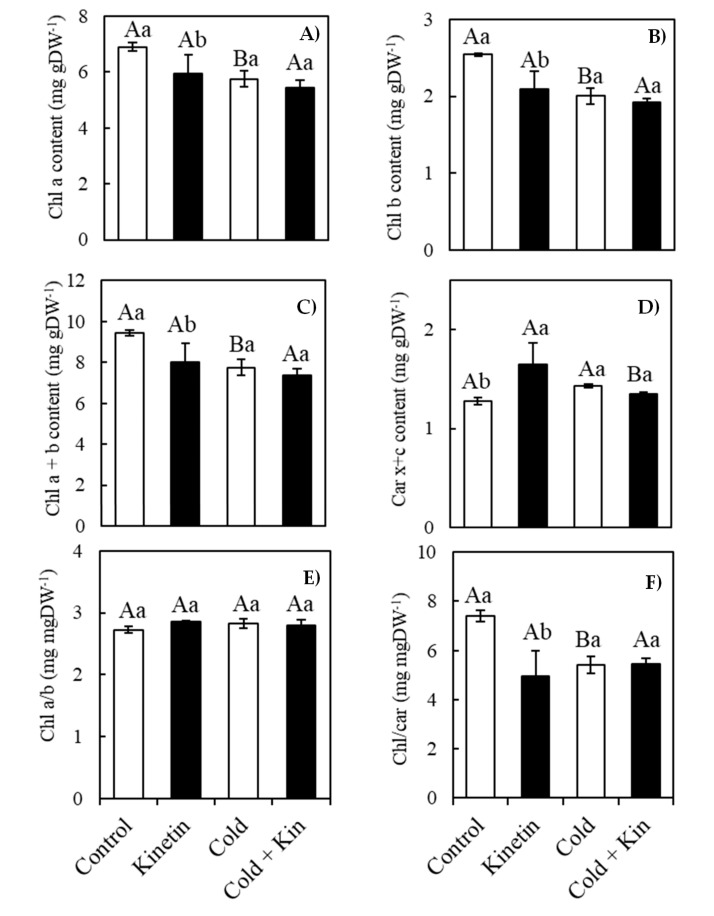
The effect of exogenous kinetin on the contents of chlorophyll a (**A**), chlorophyll b (**B**), total chlorophylls (**C**), total carotenoids (**D**) as well as the ratios of chlorophyll a/b (**E**) and total chlorophyll /total carotenoid (**F**). Kin, kinetin. Different capital letters, (A, B) denote statistically significant differences between the parameter means within temperature treatments, whereas different small letters, (a, b) denote statistically significant differences for the parameter means within kinetin treatments (*p* < 0.05, Newman Keuls’ test). Means are a replicate of three independent samples (*n* = 3). Error bars represent the standard deviation of the mean.

**Figure 3 plants-09-00281-f003:**
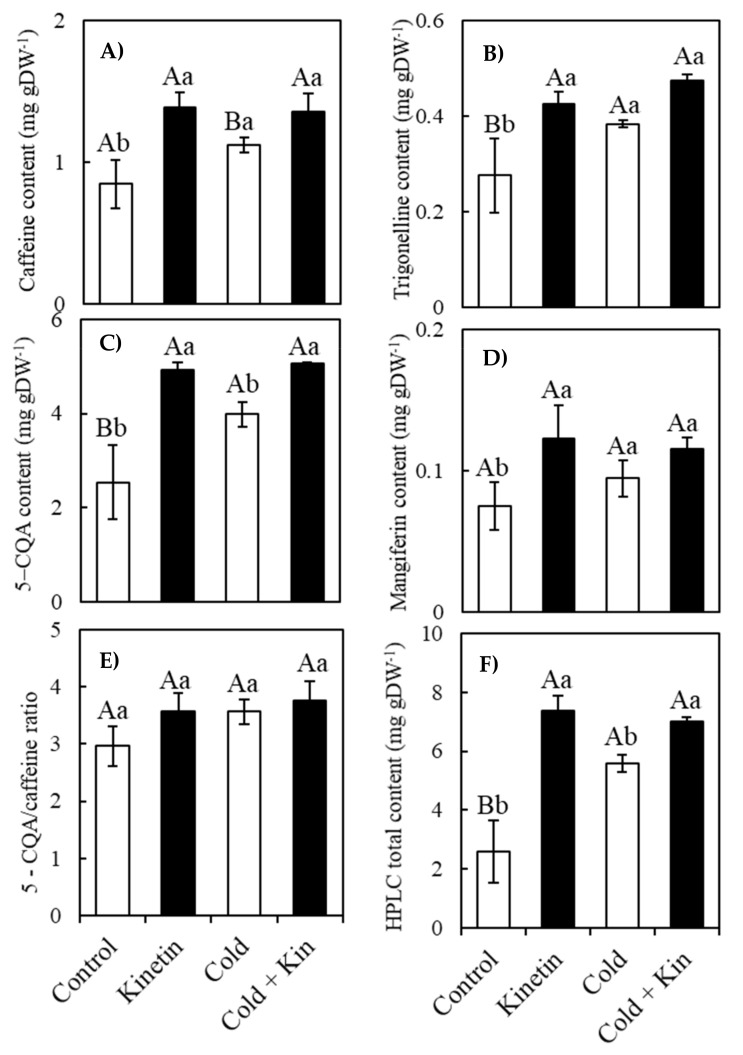
Effect of exogenous kinetin on the content of the nonenzymatic antioxidant compounds in the leaves of coffee plants under optimum and cold stress conditions. Caffeine (**A**) trigonelline (**B**); 5-caffeoylquinic acid (**C**); mangiferin (**D**); 5-caffeoylquinic acid/caffeine ratio (**E**) and HPLC total content (**F**). Kin, kinetin. Different capital letters, (A, B) denote statistically significant differences between the parameter means within temperature treatments, whereas different small letters, (a, b) denote statistically significant differences for the parameter means within kinetin treatments (*p* < 0.05, Newman–Keuls test). Means are a replicate of three independent samples (*n* = 3). Error bars represent the standard deviation of the mean.

**Figure 4 plants-09-00281-f004:**
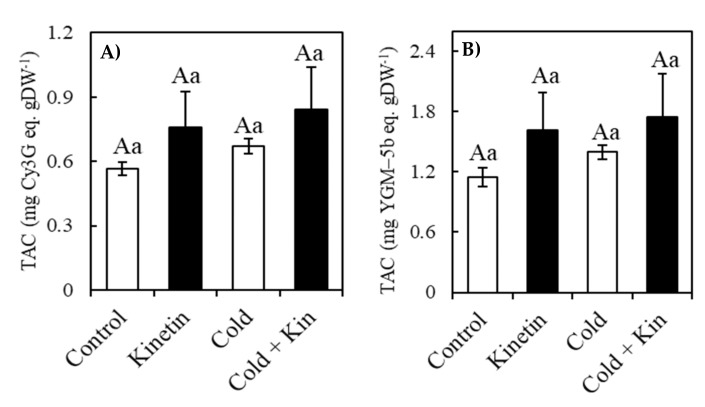
Effect of exogenous kinetin on total anthocyanin content (TAC) expressed on the basis of both peonidin-3-(6-caffeoyl-sophoride)-glucoside (**A**) and cyanidin 3 glucoside (**B**) in the leaves of the coffee plants under optimum and cold stress conditions. Kin, kinetin. Treatment means followed by the same capital letter, (A) are not statistically significantly different between the temperature treatments, whereas treatment means followed by the same small letter, (a) are not statistically significantly different between the kinetin treatments (*p* < 0.05, Newman–Keuls test). Means are a replicate of three independent samples (*n* = 3). Error bars represent standard deviation of the mean.

**Figure 5 plants-09-00281-f005:**
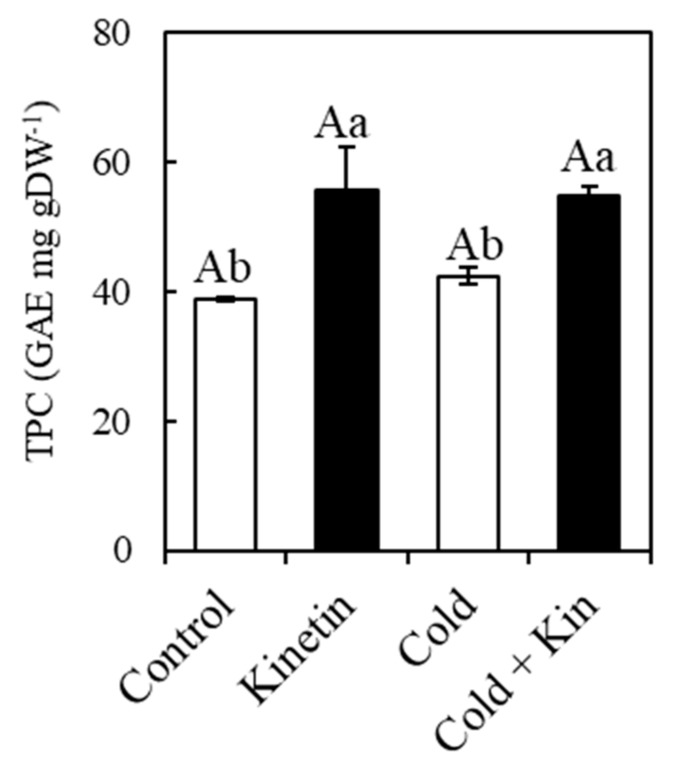
Effect of exogenous kinetin on total phenolic content (TPC) expressed on the basis of gallic acid equivalent (GAE) in the leaves of coffee plants under optimum and cold stress conditions. Kin, kinetin. Different capital letters denote the statistically significant differences between the parameter means within temperature treatments, whereas different small letters denote statistically significant differences for the parameter means within kinetin treatments (*p* < 0.05, Newman–Keuls test). Means are a replicate of three independent samples (*n* = 3). Error bars represent the standard deviation of the mean.

**Table 1 plants-09-00281-t001:** Effect of exogenous kinetin on the net CO_2_ assimilation rate (A), stomatal conductance (g_s_), intrinsic water-use efficiency (A/g_s_), instantaneous water-use efficiency (A/E), transpiration rate (TrmmoL), intracellular CO_2_ concentration (C_i_), internal to ambient CO_2_ concentration ratio (C_i_/C_a_) and mesophyll efficiency in the leaves of coffee seedlings under optimum and cold stress conditions, 25/20 °C and 12/12 °C, respectively.

Parameter	Optimum Conditions		Cold Stress Conditions	
	Control	Kinetin	Cold	Cold + Kinetin
A (µmoL CO_2_ m^−2^ s^−1^)	2.84 ± 1.61 ^Aa^	1.83 ± 0.11 ^Aa^	1.55 ± 0.55 ^Aa^	2.54 ± 0.17 ^Aa^
g_s_ (mmoL m^−2^ s^−1^)	31.9 ± 16.1 ^Aa^	17.6 ± 3.2 ^Aa^	8.12 ± 4.7 ^Ba^	18.0 ± 1.8 ^Aa^
A/g_s_ (µmoL mmol^−1^)	0.10 ± 0.04 ^Ba^	0.11 ± 0.02 ^Aa^	0.21 ± 0.05 ^Aa^	0.14 ± 0.02 ^Ab^
A/E (µmoL mmol^−1^)	4.13 ± 1.86 ^Ba^	3.99 ± 0.88 ^Aa^	10.4 ± 2.62 ^Aa^	7.03 ± 0.79 ^Ab^
Trmmol (mmoL H_2_O m^−2^ s^−1^)	0.75 ± 0.38 ^Aa^	0.47 ± 0.08 ^Aa^	0.17 ± 0.10 ^Bb^	0.36 ± 0.04 ^Aa^
c_i_ (µL L^−1^)	333 ± 125 ^Aa^	215 ± 37.8 ^Aa^	81.2 ± 54.9 ^Ba^	161.9 ± 29.7 ^Aa^
c_i_/c_a_	0.83 ± 0.31 ^Aa^	0.54 ± 0.09 ^Aa^	0.05 ± 0.003 ^Bb^	0.41 ± 0.07 ^Aa^
c_i_/g_s_	7.91 ± 1.03 ^Ab^	12.23 ± 0.15 ^Aa^	7.54 ± 3.00 ^Aa^	8.95 ± 1.04 ^Ba^

Within a row, different capital letters, (A, B) denote the statistically significant differences between the parameter means within temperature treatments, whereas different small letters, (a, b) denote the statistically significant differences for the parameter means within kinetin treatments (*p* < 0.05, Newman–Keuls test). Data are expressed as means ± S.D (*n* = 3).

**Table 2 plants-09-00281-t002:** Effect of exogenous kinetin on the PSII operating efficiency (ΦPSII), electron transfer rate (ETR), photochemical quenching (qP), non-photochemical quenching (NPQ), fluorescence quenching (qN), and the fraction of open PSII centers (qL) in the leaves of coffee seedlings under optimum and cold stress conditions, 25/20 °C and 12/12 °C, respectively.

Parameter	Optimum Condition		Cold Stress Conditions	
	Control	Kinetin	Cold	Cold + Kinetin
Φ_PSII_	0.15 ± 0.03 ^Aa^	0.13 ± 0.02 ^Aa^	0.09 ± 0.02 ^Aa^	0.09 ± 0.01 ^Aa^
ETR (µmoL m^−2^ s^−1^)	31.8 ± 6.0 ^Aa^	28.7 ± 3.6 ^Aa^	19.4 ± 4.1 ^Ba^	20.0 ± 2.9 ^Aa^
qP	0.28 ± 0.04 ^Aa^	0.24 ± 0.05 ^Aa^	0.21 ± 0.02 ^Aa^	0.23 ± 0.04 ^Ba^
qN	2.08 ± 0.01 ^Aa^	2.30 ± 0.33 ^Aa^	1.72 ± 0.15 ^Aa^	1.69 ± 0.14 ^Ba^
NPQ	1.27 ± 0.45 ^Aa^	1.12 ± 0.50 ^Aa^	0.68 ± 0.48 ^Aa^	0.71 ± 0.41 ^Aa^
qL	0.50 ± 0.08 ^Aa^	0.47 ± 0.04 ^Aa^	0.33 ± 0.05 ^Ba^	0.35 ± 0.04 ^Ba^

Within a row, different capital letters, (A, B) denote statistically significant differences between the parameter means within temperature treatments, whereas treatment means followed by the same small letter, (a) are not statistically significantly different between the kinetin treatments (*p* < 0.05, Newman–Keuls test). Data are expressed as means ± S.D (*n* = 3).

**Table 3 plants-09-00281-t003:** The effect of exogenous kinetin on maximum and effective quantum efficiency, expressed as ratios of variable to maximum and variable to initial PSII quantum efficiency in the light and in the dark-adapted states in the leaves of coffee seedlings under optimum and cold stress conditions.

Parameter	Optimum Conditions		Cold Stress Conditions	
	Control	Kinetin	Cold	Cold + Kinetin
Fv’/Fm’	0.52 ± 0.02 ^Aa^	0.56 ± 0.06 ^Aa^	0.42 ± 0.05 ^Ba^	0.40 ± 0.05 ^Ba^
Fv/Fm	0.75 ± 0.04 ^Aa^	0.75 ± 0.02 ^Aa^	0.57 ± 0.14 ^Aa^	0.58 ± 0.05 ^Aa^
Fv’/Fo’	1.08 ± 0.10 ^Aa^	1.30 ± 0.33 ^Aa^	0.72 ± 0.15 ^Aa^	0.69 ± 0.14 ^Ba^
Fv/Fo	3.02 ± 0.71 ^Aa^	3.00 ± 0.25 ^Aa^	1.46 ± 0.75 ^Ba^	1.43 ± 0.31 ^Ba^

Within a row, different capital letters, (A, B) denote statistically significant differences between the parameter means within temperature treatments, whereas treatment means followed by the same small letter, (a) are not statistically significantly different between the kinetin treatments (*p* < 0.05, Newman Keuls’ test). Data are expressed as means ± S.D (*n* = 3).

**Table 4 plants-09-00281-t004:** Effect of exogenous kinetin on the antioxidant activities of coffee leaves under optimum and cold stress conditions.

Parameter	Optimum Condition		Cold Stress Conditions	
	Control	Kinetin	Cold	Cold + Kinetin
DPPH IC_50_ µgDW mL^−1^	125.8 ± 21.5 ^Ba^	114.4 ± 12.7 ^Aa^	154.0 ± 8.5 ^Aa^	1102.8 ± 7.2 ^Ab^
DPPH TEAC (µmoL Trolox gDW^−1^)	211.3 ± 0.1 ^Ab^	317.9 ± 30.5 ^Aa^	190.2 ± 10.6 ^Ab^	270.1 ± 35. 2 ^Ba^
ABTS TEAC (µmoL Trolox gDW^−1^)	262.2 ± 3.0 ^Aa^	277.2 ± 19.4 ^Aa^	236.9 ± 23.3 ^Ab^	281.6 ± 7.15 ^Aa^
FRAP TEAC (µmoL Trolox gDW^−1^)	752.5 ± 73.8 ^Ab^	1138.6 ± 169.6 ^Aa^	542.1 ± 48.4 ^Bb^	938.7 ± 78.6 ^Ba^

Within a row, different capital letters, (A, B) denote statistically significant differences between the parameter means within temperature treatments, whereas different small letters, (a, b) denote the statistically significant differences for the parameter means within kinetin treatments (*p* < 0.05, Newman–Keuls test). Data are expressed as means ± S.D (*n* = 3).

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
