# Peer review of "Exogenous Kinetin Promotes the Nonenzymatic Antioxidant System and Photosynthetic Activity of Coffee (Coffea arabica L.) Plants Under Cold Stress Conditions"

_plants, 2020, doi:10.3390/plants9020281_

Round 1

Reviewer 1 Report

Overall paper:

The paper is scientifically sound. The methods used show good support for the analyses of several parameters. The topic is a novelty and offers a new perspective on the control of ROS under cold stress condition with rescuing by kinetin. It would be interesting to take knowledge from this paper to investigate the suitability of kinetin applications during other abiotic stresses as well.

English editing: Minor grammatical editing is required, particularly for the errors:

Line 64: grammar tense “photosynthetic apparatuses are stressed”

Line 65: “over production” is one word

Line 71: use “ROS” instead of “Reactive species”

Line 92: need space before citation

Line 148: period at end of sentence needed

Line 263: space after 50.0

Line 317/444: Kinetin should not be capitalized

Line 427: abbreviate ROS

Line 539: space between 400ppm

Introduction:

Can you add 1-2 sentence to elaborate on why/the mechanism how C. arabica is more tolerant than C. canephora? (i.e. due to genetic differences, better photosynthesis, etc.)

I like the inclusion of the supplemental table showing the chemical structures of specified analytes. Minor revision is required for alignments of the table, so that all text is centered equally and the chemical structures (i.e. gallic acid) are not compressed horizontally. What is explanation for the OH groups colored red in YGM – 5b?

Results:

Results are sufficiently explained.

Fig S1: Should the x- and y-axis be labeled?

Fig 2: Significance letters formatting needs to be adjusted to same spacing.

Fig 5: “ns” placement not consistent with Fig 4.

Discussion:

Lines 428-432: Perhaps ref 13 “ROS are good” could also be referenced here, based on concept of a balance between ROS production and scavenging.

There is a good explanation for why the specific concentrating of kinetin was studied. This is not necessary for acceptance, but it would have been nice to see a kinetin dose-dependent effect on one of the parameters under cold conditions to support that the concentration picked for the study is the most optimal.

Reviewer 2 Report

In this study, the authors investigated the effect of cytokinin on coffee plants under cold stress conditions. Here are some major concerns before the work can be published. Some examples are:

1) Abstract needs some work. It is not a good way to put detailed methods, results, conclusions all in the abstract. Please be concise.

2) For the tables in the manuscript: It seems that the author is comparing cold vs control; optimum+Kin vs cold+kin. This comparison tells us what the difference is between application of kin under cold and under optimum. However, what the readers really want to know is that cold stress causes some change to the plants and whether application of kin will make it better or worse. Therefore, a more proper way to compare is cold vs cold + kin, not the optimum+Kin vs cold+kin. As far as I can see in the tables, there is no significant difference between cold and cold+kin, except for A/gs and A/E in Table 1. This means, under cold conditions, whether you apply kin or not won't make a difference to most of the parameters. Please make better explanations in the manuscript.

3) In the results, it is better to add a conclusion after every result. Add something like "This indicates/suggests/shows that ... ". Readers need to know what point the authors are trying to make by describing this result.

4) Lines 277-278: "...treatment elicited more 5–CQA than caffeine resulting in to an increment of 20.3% compared to optimum conditions (Figure 3E)." But the figure 3E shows no difference (if ns means no significance). Why use "ns" instead of letters "abcd"?

Reviewer 3 Report

My decision for the manuscript is "major revision" and my comment is attached

Reviewer 4 Report

Manuscript “Exogenous kinetin promotes the non-enzymatic antioxidant system and photosynthetic activity of coffee (Coffea arabica L.) plants under cold stress conditions”, having importance to the concerned areas, however manuscript present new input in this field, points as mentioned below are required improvements.

Line 24-27- Needs re-writing and English improvement

Line-39-40-Sentence should not be part of abstract

 Line 92- Space before ‘[17,18]’

Line 142- What actually authors want to say in the sentence ‘(59.8%) was 59.8% higher in’ is not clear.

Table 1: Why standard deviation of may parameters under different treatments are so high. Additionally, it seems the letter of significance assigned to many means is not justified, for example in the case of gs how value 31.9 could not significantly differ with the value 17.6. Similar cases are also present with many parameters means in this table.

Similar things should also be answered for Table 2,  3 and 4 also.

In table footnotes sentence “Treatment means followed by the same letters are not statistically significant” should be “Treatment means followed by the same letters are not statistically significantly different”.

Figure 2E and 3E- Instead of writing “ns” it is better to assign the letter ‘a’ to each bar.

Figure 4- Assign letter of significant rather than writing “ns”.

Figure 5- Assign letter of significant rather than writing “ns”.

Line 401- Sentence “This is an agreement” should ‘This is in agreement’, same should be followed for the entire discussion section.

Line 498-499- Sentence needs re-writing.

Line 521- Add space between  0.35 and mM’.

Line 539- Add space.

Round 2

Reviewer 2 Report

The authors have answered all reviewers' questions and the manuscript is ready for publication.